# Early childhood trajectories of domain-specific developmental delay and gestational age at birth: An analysis of the All Our Families cohort

Nikki L. Stephenson [1,2]*, Suzanne Tough[1,3], Tyler Williamson [1], Sheila McDonald[1,3], Carly McMorrris[4], Amy Metcalfe[1,2,5]

1 Department of Community Health Sciences, Cumming School of Medicine, University of Calgary, Calgary, Alberta, Canada, 2 Department Obstetrics and Gynecology, Cumming School of Medicine, University of Calgary, Calgary, Alberta, Canada, 3 Department of Paediatrics Cumming School of Medicine, University of Calgary, Calgary, Alberta, Canada, 4 School and Applied Child Psychology, Werklund School of Education, University of Calgary, Calgary, Alberta, Canada, 5 Department of Medicine, Cumming School of Medicine, University of Calgary, Calgary, Alberta, Canada

* nstephe@ucalgary.ca

## Abstract

### Objective

To describe developmental domain-specific trajectories from ages 1 through 5 years and to estimate the association of trajectory group membership with gestational age for children born between ≥34 and <41 weeks gestation.

### Methods

Using data from the All Our Families cohort, trajectories of the domain-specific Ages & Stages Questionnaire scores were identified and described using group-based trajectory modeling for children born ≥34 and <41 weeks of gestation (n = 2664). The trajectory groups association with gestational age was estimated using multinomial logistic regression.

### Results

Across the five domains, 4–5 trajectory groups were identified, and most children experienced changing levels of risk for delay over time. Decreasing gestational age increases the Relative risk of delays in fine motor (emerging high risk: 1.46, 95% CI: 1.19–1.80; resolving moderate risk: 1.11, 95% CI: 1.03–1.21) and gross motor (resolving high risk: 1.21, 95% CI: 1.04–1.42; and consistent high risk: 1.64, 95% CI: 1.20–2.24) and problem solving (consistent high risk: 1.58 (1.09–2.28) trajectory groups compared to the consistent low risk trajectory groups.

**Data Availability Statement:** Data that support the findings of this study are available from the All Our Families Cohort (https://ucalgary.ca/allourfamilies),

and metadata of variables can be found at https://www.maelstrom-research.org/study/aof. Restrictions apply to the availability of these deidentified data, which were used according to data sharing agreements.

**Funding:** NS was funded throughout this work by the Alberta Children's Hospital Research Institute Graduate Scholarship, the Alberta Graduate Excellence Scholarship, the Faculty of Graduate Studies Doctoral Scholarship, and the University of Calgary Graduate Studies Scholarship. The All Our Families cohort was funded through the Alberta Innovates Interdisciplinary Team Grant #200700595, the Alberta Children's Hospital Foundation, and the MaxBell Foundation. The funders had no role in study design, data collection and analysis, decision to publish, or preparation of the manuscript.

**Competing interests:** The authors have declared that no competing interests exist.

## Conclusion

This study highlights the importance of longitudinal analysis in understanding developmental processes; most children experienced changing levels of risk of domain-specific delay over time instead of having a consistent low risk pattern. Gestational age had differential effects on the individual developmental domains after adjustment for social, demographic and health factors, indicating a potential role of these factors on trajectory group membership.

## Introduction

Gestational age exists along a continuum, and the risk of developmental delays has an inverse dose-response association with gestational age [1,2]. At a population level, the gestational age distribution at birth has shifted over the last 40 years, with the mean gestational age decreasing from 40 to 39 weeks [3,4]. Though adverse developmental outcomes are more severe and best established with infants born extremely preterm, the increasing numbers of infants born between $\geq 34$ and $< 39$ weeks may result in a higher total number of children at risk in the population [5]. A growing body of literature demonstrates this potential gradient of risk for developmental delays, neurodevelopmental disabilities, cognitive and academic difficulties, behavioral and emotional challenges across the gestational age continuum [6–10].

The most receptive stage of development is during early childhood; during this period, there is increased potential to permanently alter developmental trajectories and minimize disadvantages from accumulating through intervention [11]. Many interventions have sought to improve outcomes for infants born before 34 weeks gestation [12], but few interventions have been developed that target or include infants >34 weeks [8]. Understanding the short and long-term risk associated along the continuum of gestational age is vital to developing early childhood interventions for developmental delay [8]. However, few studies have examined longitudinal development patterns or how those patterns of development differ across domains [1,2,13–15]. The objectives of the present study were to 1) describe developmental domain-specific trajectories as measured by the Ages & Stages Questionnaire 3rd edition (ASQ-3) from 1 to 5 years of age, and 2) to estimate the association of trajectory group membership with gestational age for children born between $\geq 34$ and $< 41$ weeks gestation.

## Methods

### Study design

This observational study uses data from the All Our Families (AOF) study, a community-based prospective pregnancy cohort that recruited 3387 pregnant women via prenatal serology testing, primary healthcare clinics, and community posters in Calgary, Alberta, from May 2008 until May 2011 [16]. Details of AOF cohort methods, including study design, setting, and participant selection/eligibility/follow-up, have been reported elsewhere, and persons were eligible if they were $\geq 18$ years of age, accessing prenatal care in Calgary, could complete questionnaires using English, and their pregnancy was $\leq 25$ completed weeks gestation [17]. Demographically, the AOF cohort is broadly representative of the pregnant and parenting population of urban centers in Canada [16].

Participants were initially invited to complete three questionnaires at $\leq 25$ weeks pregnant, 34–36 weeks pregnant, and 4 months post-partum and consented to linkage to labor and

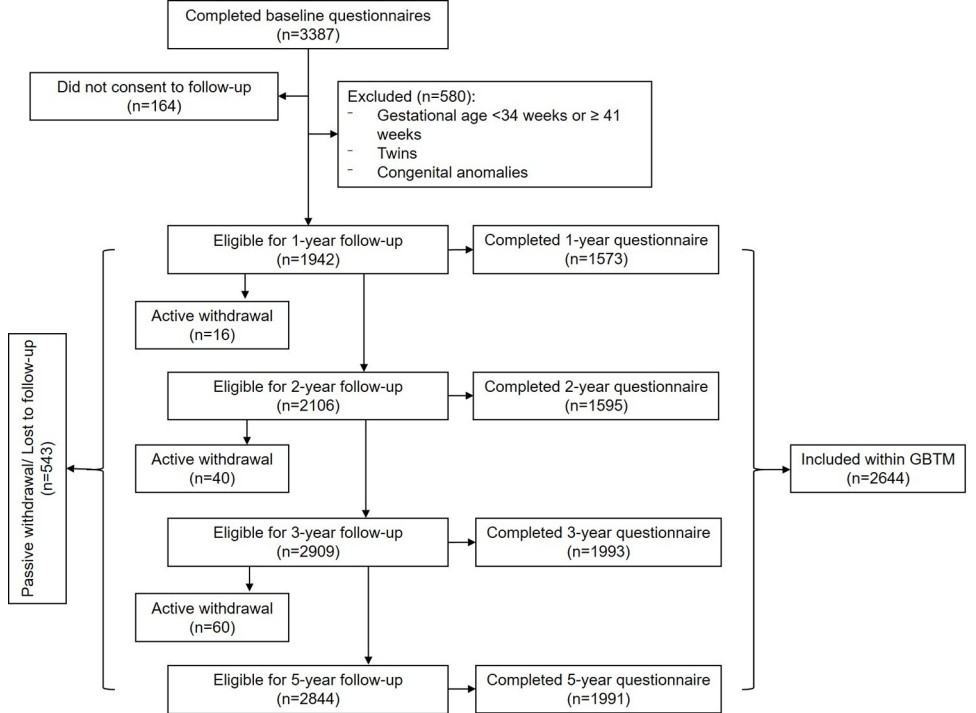

**Fig 1. Study eligibility criteria and sample size.** n: The number of participants, GBTM: Group-based trajectory modeling.

delivery health records [16,18]. Annual follow-up questionnaires were sent out on the child's birthdate to those participants who consented to be contacted for future research [16]. Some follow-up questionnaires were delayed to secure funding and ethical approval, resulting in participants becoming ineligible as their children aged out of the questionnaire's administrative window (Fig 1) [16,19]. Additional inclusion criteria for the current study (n = 2644) were participant dyads ≥18 years of maternal age who speak English with singleton pregnancies and children born ≥34 and <41 weeks of gestation with no congenital anomalies (Fig 1).

## Participant consent and ethics statement

Cohort participants were informed of and agreed to (via signature of consent forms) the known research intent, risks, benefits, and privacy considerations during initial recruitment at each follow-up questionnaire for themselves and their child [16]. Participants included in the current study also agreed, via the same informed consent processes, for their de-identified research data to be stored and used for secondary data analysis subsequent to ethical review. The present study, using de-identified participant data, was approved by the Conjoint Health Research and Ethics Board at the University of Calgary (REB21-0249).

## Exposure

The exposure of interest, gestational age at birth, was derived from early ultrasound dating within electronic medical records (EMR) or maternal reported last menstrual period if EMR data were unavailable. EMR gestational age was collected for 85% of participants in this study. Previous analysis using this cohort data showed high validity for the maternal report of gestational age at birth and labor and delivery outcomes [18]. This study included only those born

≥34 and <41 weeks gestational age, and gestational age was reverse coded with 40 weeks as a baseline for modeling.

## Outcome

Child developmental outcomes were measured using the ASQ-3, a caregiver-completed screening tool [20]. The ASQ-3 was administered at 1, 2, 3, and 5 years of age [16,20]. The ASQ-3 contains 30 items divided into five domains of development: communication, problem solving, personal social, fine motor, and gross motor. Each item is assigned a point value to generate a domain-specific score out of 60, with higher scores indicating a lower risk of delay within the specific domain. Scores 1 standard deviation (SD) below the United States (US) normalized mean suggest that the child is at risk of delay and should be monitored, and scores ≥2 SD below indicate that the child is at risk of delay and should be further assessed [20].

## Potential confounding variables

Various perinatal sample characteristics and factors influencing child development were assessed as confounding variables. Maternal age at delivery (years), maternal education (≤high school or >high school), maternal self-identified ethnicity (dichotomised into White or BIPOC [Black, Indigenous, People of Color] due to sample size limitations), parity (primiparous or multiparous) are self-reported variables. Antenatal anxiety (≥40 on the Spielberg State Anxiety scale) [21], antenatal depression (≥10 on the Edinburgh Postpartum Depression Scale) [22], and antenatal stress (≥19 on the Perceived Stress Scale, which is 1 SD above the mean [23]) variables were collected via validated maternal self-report screening tools during the second trimester. Binge drinking (defined as consuming ≥5 alcoholic drinks), smoking (cigarettes), and drug use (defined as broadly as street drugs, with no differentiation for cannabis) were self-reported variables assessed for the entirety of pregnancy, including before pregnancy recognition. Child sex (male/female), small for gestational age (SGA; birthweight below 10th percentile of sex-specific birthweight as a proxy for intrauterine growth restriction [24]), and neonatal intensive care unit admission (NICU; a proxy for birthing related morbidities [25]) variables originated from the newborn discharge abstract of the EMR.

## Statistical analysis

All analyses used Stata 16 IC software [26]. Descriptive analysis presents the distribution of sample characteristics and factors influencing child development for the full sample, and across the range of gestational ages via stratification by gestational age categories. These gestational age categories are: late preterm (LPT: ≥34 and <37 weeks gestation), early term (ET: ≥37 and <39 weeks gestation), and full term (FT: >39 and <41 weeks gestation).

Trajectories on ASQ-3 domain-specific scores between 1 and 5 years of age were examined using group-based trajectory modeling (GBTM). GBTM applies finite mixture modeling using a maximum likelihood estimation, combining single-group models within a multiple-group model structure to identify clusters of individuals with similar trajectories [27,28]. This method estimates the probability of trajectory group membership used in subsequent regression [28]. We considered each child's age at ASQ-3 administration at 1, 2, 3, and 5 years as time 0, 1, 2, and 3. Domain-specific ASQ-3 scores were modeled separately within the TRAJ STATA plug-in using a censored normal model to account for clustering at either end of the scale [29]. Model selection used both the Bayesian Information Criterion and the Akaike Information Criterion for base specification [30], followed by altering the polynomial orders of the trajectories using the Wald z-test and visualization of the trajectories to determine the most parsimonious number of groups [28]. The number of trajectory groups was also visually

assessed via graphing the mean outcome values and confidence intervals throughout the model selection process to ensure that distinct trajectory groups were not combined or separated based solely on Bayesian Information Criterion and the Akaike Information Criterion values. Model adequacy was determined using: 1) average posterior probability of assignment for all groups, 2) odds of correct classification, and 3) probability of group membership [28]. Proportions within each domain-specific trajectory group that scored 1 SD and 2 SD below the normalized mean at least once from 1 to 5 years of age were used to describe the level of risk of developmental delay among trajectory groups [20]. Proportions within each domain-specific trajectory group that scored 1 SD below the normalized mean at each assessment point were used to characterize the trajectory groups' transitions between risk levels. "Emerging" described an increase in risk over time, "resolving" described a decrease in risk over time, "consistent" described relatively constant risk over time, and "transient" described changing risk, which may be both increasing or decreasing over time but was not consistent.

Once the most parsimonious GBTM model had been fit for each domain-specific outcome, a multinomial logistic regression within the TRAJ plug-in estimated the association between gestational age and posterior probabilities of trajectory group membership using starting values [28]. Incorporation of multinomial logistic regression within the trajectory model makes the probability of group membership a function of the covariates, allowing accurate estimation of the association of gestational age with group membership probability [28]. Risk ratios (RR) were generated by appending observations with no outcome data to generate group membership probabilities based on the inputted TRAJ risk factor settings, and 95% confidence intervals (CI) were generated through parametric bootstrapping [28]. Informed by crosstabulation of potential confounding factors and trajectory groups, the inclusion of confounding variables was determined through backward elimination.

## Missing data

The dataset has no missing data for the exposure (gestational age), and GBTM accommodates missing outcome data due to intermittent or missed assessments. However, missing data within potential confounding variables does have the potential to bias the estimate of the association between gestational age and the probability of group membership. To address missing data, the percentage of missing data for all potential confounding variables was computed during the descriptive analysis. Providing that the fitted models did not include variables with greater than 5% data missingness [31], a complete cases approach is used to handle missing data.

## Results

### Descriptive analysis

Overall, 2,644 children were included in this analysis, of which 6.1% were LPT, 29.7% were ET, and 64.2% were FT. Children born LPT were significantly more likely to be admitted to the NICU (neonatal intensive care unit) than infants born either ET or FT, and their mothers were significantly more likely to have experienced depressive symptoms during pregnancy when compared to infants born FT. No other significant differences (via exclusivity of presented confidence intervals) were noted in the distribution of sample characteristics between LPT, ET, and FT born infants (Table 1).

### Description of trajectory groups by model

A 4-group model emerged as the best fit for the communication, personal social, and fine motor domains, while a 5-group model was used for gross motor and problem solving. Further

**Table 1. Sample characteristics stratified by gestational age category.**

| Variable | | | Percent of data missingness (n = 2644) % (n) | Full sample (n = 2644) % (95% CI) | Gestational age category | | |
|---|---|---|---|---|---|---|---|
| | | | | | LPT (n = 158) % (95% CI) | ET (n = 787) % (95% CI) | FT (n = 1699) % (95% CI) |
| Child | Child sex[a] | Male | 0.0 (n = 0) | 52.2 (50.3–54.1) | 61.4 (53.6–68.7) | 51.5 (48.0–54.9) | 51.7 (49.3–54.0) |
| | SGA | Yes | 10.2 (n = 271) | 10.9 (9.7–12.2) | 10.6 (6.5–16.8) | 9.5 (7.5–11.9) | 11.5 (10.0–13.2) |
| | NICU admission | Yes | 8.9 (n = 234) | 3.6 (2.9–4.4) | 23.6 (17.4–31.2)[b] | 3.3 (2.3–4.9) | 1.8 (1.3–2.6) |
| Maternal | Age[a] | ≥35 years | 0.6 (n = 16) | 20.3 (18.8–21.9) | 19.7 (14.2–26.8) | 21.5 (18.7–24.5) | 19.9 (18.0–21.9) |
| | Education | ≤High school | 0.8 (n = 22) | 10.7 (9.6–12.0) | 9.0 (5.4–14.7) | 11.4 (9.3–13.8) | 10.6 (9.2–12.1) |
| | Household income | <$40,000 | 0.0 (n = 0) | 8.3 (7.3–9.4) | 9.3 (5.6–15.1) | 8.3 (6.6–10.5) | 8.1 (6.9–9.6) |
| | Ethnicity[a] | BIPOC | 0.0 (n = 0) | 22.4 (20.8–24.0) | 29.1 (22.6–36.7) | 24.7 (21.8–27.8) | 20.7 (18.8–22.7) |
| | Parity | Primiparous | 1.3 (n = 37) | 46.4 (44.5–48.3) | 54.3 (46.3–62.1) | 43.2 (39.8–46.8) | 47.2 (44.8–49.6) |
| Pregnancy | Anxiety[a] | Yes | 4.2 (n = 111) | 15.6 (14.3–17.1) | 21.6 (15.7–29.0) | 16.4 (13.9–19.2) | 14.7 (13.1–16.5) |
| | Depression[a] | Yes | 1.1 (n = 29) | 7.6 (6.7–8.7) | 14.3 (9.6–20.7)[b] | 7.9 (6.2–10.0) | 6.9 (5.8–8.3) |
| | Stress | Yes | 1.7 (n = 47) | 16.9 (15.5–18.4) | 19.5 (14.0–26.5) | 16.2 (13.8–19.0) | 16.9 (15.2–18.8) |
| | Binge drinking | Yes | 10.4 (n = 275) | 19.5 (17.4–21.8) | 21.5 (13.2–33.2) | 19.1 (15.3–23.6) | 19.5 (16.9–22.3) |
| | Smoking | Yes | 10.5 (n = 277) | 11.7 (10.4–13.0) | 18.2 (12.5–25.7) | 11.8 (9.6–14.4) | 11.1 (9.6–12.7) |
| | Drug use | Yes | 3.6 (n = 96) | 3.9 (3.2–4.7) | 3.4 (1.4–8.0) | 4.5 (3.2–6.3) | 3.6 (2.8–4.7) |

LPT: Late preterm, ET: Early term, FT: Full term, CI: Confidence interval, BIPOC: Black, Indigenous, People of Colour, SGA: Small for gestational age; NICU: Neonatal intensive care.

[a] denotes those confounding variables included in the fitted multinomial logistic regression models, percent of data missing among all variables included in the models are 4.6% (123/2644).

[b] denotes statistically significant difference as the 95% CI of the proportions are exclusive among gestational age categories.

details on model adequacy are presented in Table 2. These identified trajectory groups are graphically represented by line graphs of the of mean ASQ-3 scores from ages 1 though 5 in Fig 2.

Description of the sample sizes, polynomial order, slope direction(s), and levels and patterns of risk for these trajectory groups are detailed in Table 3. Three levels of risk were defined when considering the proportion of children within each trajectory group who score 1 SD below the corresponding domain-specific US normalized mean at least once before 5 years of age: low (below 10%), moderate (10–49%) and high (50–100%). Patterns of risk, characterized using the proportions of scores 1 SD below the normalized mean at each assessment point, describe how the trajectory groups move between levels of risk, are shown within Table 4.

Table 3 shows that within the communication domain, most of the sample (64.6%; 95% CI:62.8–66.4%) is identified as belonging within trajectory group 4. The proportion of children who score 1 SD below the domain-specific US normalized mean is 0.8% (0.4–1.5%); therefore, within the communication domain, trajectory group 4 is described as having a low level of risk of developmental delay (Table 3). Table 4 shows that 0.2% (0.0–1.3), of the children within trajectory group 4 score 1 SD below the domain-specific US normalized mean at 1 year, and 0.0%

**Table 2. Measures of model adequacy per ASQ-3 (ages & stages questionnaire 3rd edition) domain specific model.**

| Model | N1 | BIC (N1) | N2 | BIC (N2) | AIC | Group | Polynomial order | Wald test of polynomial order | Probability of group membership | Average posterior probability of assignment | Odds of correct classification |
|---|---|---|---|---|---|---|---|---|---|---|---|
| Communication | 5033 | -13942.65 | 1888 | -13934.8 | -13890.46 | 1 | 2 | p = 0.0014 | 0.6 | 1 | 3,256 |
| | | | | | | 2 | 1 | p<0.0001 | 3.94 | 0.8 | 96 |
| | | | | | | 3 | 2 | p = 0.0007 | 47.05 | 0.8 | 4 |
| | | | | | | 4 | 3 | p<0.0001 | 48.42 | 0.7 | 2 |
| Personal social | 5044 | -13419.99 | 1888 | -13413.11 | -13374.31 | 1 | 2 | p<0.0001 | 12.5 | 0.7 | 18 |
| | | | | | | 2 | 2 | p<0.0001 | 47.03 | 0.6 | 2 |
| | | | | | | 3 | 0 | p<0.0001 | 0.81 | 0.9 | 901 |
| | | | | | | 4 | 2 | p<0.0001 | 39.66 | 0.7 | 5 |
| Fine motor | 5026 | -15430.87 | 1883 | -15423.01 | -15374.91 | 1 | 3 | p<0.0001 | 3.44 | 0.9 | 166 |
| | | | | | | 2 | 3 | p<0.0001 | 11.85 | 0.8 | 32 |
| | | | | | | 3 | 2 | p<0.0001 | 52.62 | 0.7 | 2 |
| | | | | | | 4 | 0 | p<0.0001 | 32.08 | 0.7 | 5 |
| Gross motor | 5040 | -11687.31 | 1887 | -11679.45 | -11635.11 | 1 | 3 | p<0.0001 | 10.63 | 0.9 | 57 |
| | | | | | | 2 | 0 | p<0.0001 | 0.77 | 0.8 | 552 |
| | | | | | | 3 | 0 | p<0.0001 | 19.07 | 0.6 | 7 |
| | | | | | | 4 | 2 | p<0.0001 | 44.16 | 0.6 | 2 |
| | | | | | | 5 | 1 | p<0.0001 | 25.38 | 0.7 | 8 |
| Problem solving | 5029 | -12957.12 | 1884 | -12949.76 | -12908.2 | 1 | 2 | p = 0.0155 | 0.81 | 0.9 | 753 |
| | | | | | | 2 | 0 | p<0.0001 | 8.89 | 0.7 | 25 |
| | | | | | | 3 | 1 | p<0.0001 | 69.88 | 0.7 | 1 |
| | | | | | | 4 | 1 | p<0.0001 | 20.42 | 0.7 | 8 |
| | | | | | | 5 | 1 | p<0.0001 | 21.68 | 0.6 | 6 |

N1: Number of observations across persons and time, BIC: Bayesian Information Criterion, N2: Observations in model with at least 3 measures, AIC: Akaike Information Criterion.

(0.0–0.4), 0.9% (0.4–1.9), and 0.0% (0.0–0.4) at ages 2, 3, and 4 respectively. As these proportions are relatively consistent over time, the pattern of risk for trajectory group 4 within the communication domain is described as consistent (Table 4), and trajectory group 4 is described as having a low consistent pattern of risk of developmental delay (Table 3).

Within all other domains, the majority of participants are classified as moderate risk with either a transient (personal social) or resolving (fine motor, gross motor, problem solving) pattern.

## Multinomial regression results

Table 5 shows the relative risk for the association of gestational age at birth with the probability of group membership to trajectory groups per domain-specific model. Gestational age, infant sex, maternal age, maternal ethnicity, and prenatal anxiety and depression were included as covariates in all models (see Table 6 for sample characteristics by trajectory group). Gestational age was not consistently associated with membership in high-risk trajectories. While membership in high-risk trajectory groups was often inversely associated with gestational age in crude models, these associations were attenuated following adjustment for confounders. In fully adjusted models, gestational age was inversely associated with membership in the high-risk group for problem solving, gross motor, and fine motor domains. Decreasing gestational age was not associated with an elevated risk of delay in the communication or personal social domains after adjustment for confounding.

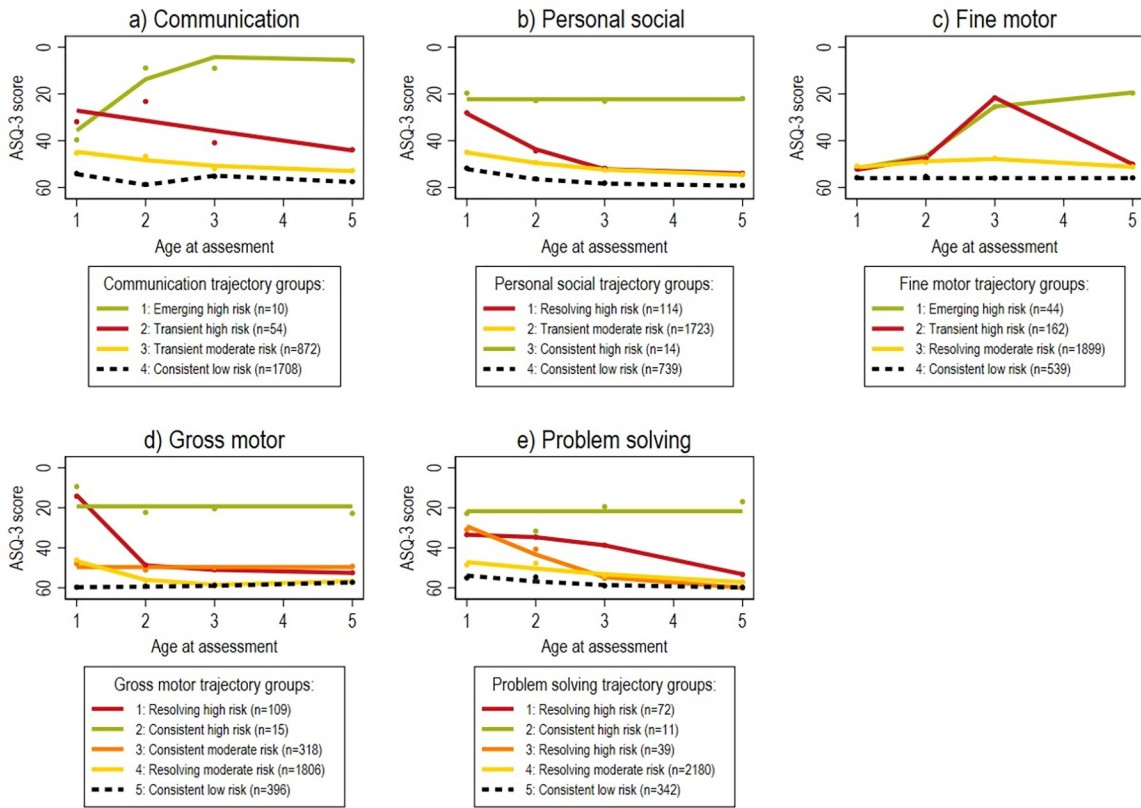

**Fig 2. Visual representation of the mean ASQ-3 scores for each trajectory group by domain.**

## Discussion

Using a large community-based prospective pregnancy cohort, this study sought to describe ASQ-3 domain-specific trajectories of child development from ages 1 through 5 years and estimate the association of trajectory group membership with gestational age for children born between ≥34 and <41 weeks, controlling for covariates. The five child development domains measured within the ASQ-3 were independently modeled to determine the number and shape of potential trajectory groups within each domain, and the trajectories were described according to the pattern and level of risk for developmental delay. This study found that the probability of group membership within high-risk groups for fine motor, gross motor, and problem solving domains of development increases with decreasing gestational age.

Our findings are consistent with previous studies showing a dose-response association between gestational age at birth and developmental delays assessed via the ASQ [1,2,14,15]. In 2015, Schonhaut et al. found an inverse dose-response association between gestational age and developmental delay using the ASQ-3, looking cross-sectionally at infants born LPT, ET, and FT [2]. In 2017, Hornman et al. found that children born preterm had increased odds of scoring below 2 SD on any domain compared to FT when using the Dutch version of the Ages & Stages Questionnaire [32]. The literature supports the idea that gestational age for those born between ≥34 and <41 weeks is inversely related to an increased risk of developmental delay. In examining children longitudinally from 4 months through 3 years of age, Hochstedler et al. also found gestational age was inversely associated with the odds of scoring below 2 SD on any ASQ domain [14]. However, literature examining the patterns of development from birth to 5

**Table 3. Description of sample sizes, polynomial order, slope descriptions, and levels and patterns of risk for trajectory groups by ASQ-3 (ages & stages questionnaire 3rd edition) domain.**

| Domain-specific model | Trajectory group number | Trajectory group (n) | Proportion of domain-specific model sample (% and 95% CI) | Polynomial order | Slope description | Proportion of trajectory group scoring 1 SD below at least once (% and 95% CI) | Proportion of trajectory group scoring 2 SD below at least once (% and 95% CI) | Risk level[a] | Risk pattern |
|---|---|---|---|---|---|---|---|---|---|
| Communication (n = 1888) | 1 | 25 | 0.4 (0.2–0.7) | 2 | Positive slope from age 1 until age 3, which plateaus but remains high until age 5 | 100% (86.3–100.0) | 100.0% (86.3–100.0) | High | Emerging |
| | 2 | 311 | 2.0 (1.6–2.7) | 1 | Negative slope from age 1 until age 5 | 67.2% (61.8–72.2) | 27.4% (22.6–32.8) | High | Transient |
| | 3 | 522 | 33.0 (31.2–34.8) | 2 | Marginally negative slope from age 1 until age 5 | 12.5% (9.9–15.6) | 1.7% (0.9–3.3) | Moderate | Transient |
| | 4 | 1030 | 64.6 (62.8–66.4) | 3 | Transient slope from age 1 through 5 | 0.8% (0.4–1.5) | 0.0% (0.0–0.4) | Low | Consistent |
| Personal social (n = 1888) | 1 | 114 | 24.6 (23.0–26.3) | 2 | Negative slope from age 1 until age 3, which plateaus but remains low through until age 5 | 100% (96.8–100.0) | 58.1% (48.5–67.1) | High | Resolving |
| | 2 | 967 | 73.4 (71.7–75.1) | 2 | Marginally negative slope from age 1 until age 5 | 27.4% (24.7–30.3) | 5.5% (4.2–7.1) | Moderate | Transient |
| | 3 | 14 | 0.5 (0.3–0.9) | 0 | No slope, but remains constant from age 1 through age 5 | 100% (76.8–100.0) | 100.0% (76.8–100.0) | High | Consistent |
| | 4 | 793 | 1.4 (1.0–2.0) | 2 | Marginally downward slope from age 1 until age 5 | 1.1% (0.6–2.2) | 0.0% (0.0–0.5) | Low | Consistent |
| Fine motor (n = 1883) | 1 | 44 | 1.7 (1.2–2.2) | 3 | Marginally positive slope from age 1 until age 2, but the slope increases from age 2 until 3 and remains positive but reduces in magnitude until age 5 | 100% (92.0–100.0) | 87.8% (73.8–94.8) | High | Emerging |
| | 2 | 162 | 6.1 (5.3–7.1) | 3 | Marginally positive slope from age 1 until age 2, but the slope increases at age 2 and while maintaining a similar magnitude the slope reverses direction at age 3 until age 5 | 100% (97.7–100.0) | 42.7% (35.0–50.7) | High | Transient |
| | 3 | 1138 | 71.8 (70.1–73.5) | 2 | Generally constant, with almost imperceptible positive slope until age 3, where it becomes negative through until age 5 | 22.0% (19.7–24.5) | 5.4% (4.2–6.9) | Moderate | Resolving |
| | 4 | 539 | 20.4 (18.9–22.0) | 0 | No slope, but remains constant from age 1 through age 5 | 1.3% (0.6–2.7) | 0.0% (0.0–3.3) | Low | Consistent |

(*Continued*)

**Table 3.** (Continued)

| Domain-specific model | Trajectory group number | Trajectory group (n) | Proportion of domain-specific model sample (% and 95% CI) | Polynomial order | Slope description | Proportion of trajectory group scoring 1 SD below at least once (% and 95% CI) | Proportion of trajectory group scoring 2 SD below at least once (% and 95% CI) | Risk level[a] | Risk pattern |
|---|---|---|---|---|---|---|---|---|---|
| **Gross motor (n = 1887)** | 1 | 109 | 4.1 (3.4–5.0) | 3 | Steep negative slope from age 1 until age 2 at which point the slope is only marginally negative | 100% (96.7–100.0) | 84.4% (76.3–90.1) | High | Resolving |
| | 2 | 15 | 0.6 (0.3–0.9) | 0 | No slope, but remains constant from age 1 through age 5 | 100% (78.2–100.0) | 100.0% (78.2–100.0) | High | Consistent |
| | 3 | 318 | 12.0 (10.8–13.3) | 0 | No slope, but remains constant from age 1 through age 5 | 68.6% (63.2–73.4) | 19.4% (15.3–24.3) | High | Transient |
| | 4 | 1049 | 68.3 (66.5–70.1) | 2 | Negative slope from age 1 until age 3, which plateaus but remains low through until age 5 | 13.7% (11.8–15.9) | 0.9% (0.5–1.7) | Moderate | Resolving |
| | 5 | 396 | 15.0 (13.7–16.4) | 1 | Minor positive slope from age 1 through 5 | 0.5% (0.1–2.0) | 0.3% (0.0–1.8) | Low | Consistent |
| **Problem solving (n = 1884)** | 1 | 72 | 2.7 (2.2–3.4) | 2 | Minor negative slope from age 1 until 3 at which point it increases in magnitude until age 5 | 100% (78.2–100.0) | 87.0% (76.8–93.1) | High | Resolving |
| | 2 | 11 | 0.4 (0.2–0.7) | 0 | No slope, but remains constant from age 1 through age 5 | 100% (71.5–100.0) | 100.0% (71.5–100.0) | High | Consistent |
| | 3 | 39 | 1.5 (1.1–2.0) | 1 | Steep negative slope from age 1 until age 3 at which point the slope decreases in magnitude | 100% (91.0–100.0) | 76.9% (61.3–87.5) | High | Resolving |
| | 4 | 1420 | 82.5 (81.0–83.9) | 1 | Minor negative slope from age 1 through 5 | 21.0% (18.9–23.2) | 2.6% (1.9–3.6) | Moderate | Resolving |
| | 5 | 342 | 12.9 (11.7–14.3) | 1 | Minor negative slope from age 1 through 5 | 0% (0.0–1.1) | 0.0% (0.0–1.1) | Low | Consistent |

n: Sample size, CI: Confidence interval, SD: Standard deviation.

[a] Three levels of risk were determined when looking at the proportion of children within each trajectory group who score 1 SD below the corresponding domain-specific US normalized mean at least once before age 5 years: Low (below 10%), moderate (49–10%) and high (100–50%).

years of age is more heterogeneous and limited. Hornman and colleagues' (2017) paper found that the pattern of developmental delay between ages 4 and 5 years was different between EPT and LPT groups and between the differing domains [32]. And, a 2018 paper, assessing ASQ-3 at 8-, 18-, and 24-months scores from children born between ≥34 and <41 weeks, found that decreasing gestational age at birth was a risk factor for boys who had scored 2 SD below the norm in 1 or 2 of the ASQ domains during that time [33]. The current study shows an inverse dose-response association between gestational age and developmental delay over the entirety of early childhood, and that the association differs by developmental domain.

To optimize child developmental outcomes through the provision of early intervention, those who fall within the consistent or emerging high risk trajectories in a specific domain may benefit from further assessment. Those within the consistent low risk trajectories describe

**Table 4. Frequency proportions of participants which score below 1 standard deviation within each ASQ-3 (ages & stages questionnaire 3rd edition) domain specific trajectory group over time.**

| Domain | Trajectory group | Trajectory group sample size (n) | 1 year % (95% CI) | 2 years % (95% CI) | 3 years % (95% CI) | 5 years % (95% CI) | Trajectory descriptor [a] |
|---|---|---|---|---|---|---|---|
| **Communication** | | | n = 1059 | n = 1187 | n = 1434 | n = 1353 | |
| | 1 | 25 | 18.2% (4.6–50.7) | 100% (86.3–100.0) | 100% (86.3–100.0) | 84.2% (60.8–94.8) | Emerging |
| | 2 | 311 | 21.9% (16.9–27.9) | 58.0% (51.4–64.2) | 19.1% (14.6–24.7) | 20.2% (15.4–26.0) | Transient |
| | 3 | 522 | 1.0% (0.3–3.0) | 4.9% (3.1–7.7) | 1.0% (0.4–2.5) | 10.2% (7.6–13.6) | Transient |
| | 4 | 1030 | 0.2% (0.0–1.3) | 0.0% (0.0–0.4) | 0.9% (0.4–1.9) | 0.0% (0.0–0.4) | Consistent |
| **Personal social** | | | n = 1058 | n = 1190 | n = 1430 | n = 1366 | |
| | 1 | 114 | 95.5% (89.5–98.1) | 47.1% (36.7–57.7) | 11.0% (5.8–19.8) | 16.2% (9.4–26.4) | Resolving |
| | 2 | 967 | 4.9% (3.3–7.1) | 21.1% (18.1–24.6) | 7.9% (6.1–10.0) | 13.6% (11.2–16.4) | Transient |
| | 3 | 14 | 100% (76.8–100.0) | 100% (76.8–100.0) | 100% (76.8–100.0) | 100% (76.8–100.0) | Consistent |
| | 4 | 793 | 1.6% (0.8–3.4) | 0.4% (0.1–1.6) | 0.0% (0.0–0.5) | 0.0% (0.0–0.5) | Consistent |
| **Fine motor** | | | n = 1059 | n = 1184 | n = 1428 | n = 1355 | |
| | 1 | 44 | 16.7% (6.4–36.9) | 37.9% (22.4–56.4) | 71.4% (54.5–83.9) | 100.0% (92.0–100.0) | Emerging |
| | 2 | 162 | 13.6% (7.7–22.9) | 19.3% (12.9–27.8) | 100.0% (97.7–100.0) | 4.3% (1.8–9.9) | Transient |
| | 3 | 1138 | 14.6% (12.0–17.5) | 14.9% (12.5–17.8) | 3.5% (2.5–5.0) | 5.6% (4.2–7.4) | Resolving |
| | 4 | 539 | 1.3% (0.5–3.3) | 0.6% (0.1–2.3) | 0.0% (0.0–0.7) | 0.2% (0.0–1.7) | Consistent |
| **Gross motor** | | | n = 1059 | n = 1187 | n = 1434 | n = 1353 | |
| | 1 | 109 | 100.0% (96.7–100.0) | 38.2% (28.7–48.7) | 31.3% (22.3–42.0) | 9.8% (5.0–18.3) | Resolving |
| | 2 | 15 | 83.3% (36.8–97.7) | 100.0% (78.2–100.0) | 100.0% (78.2–100.0) | 100.0% (78.2–100.0) | Consistent |
| | 3 | 318 | 16.9% (11.6–24.0) | 34.7% (28.4–41.5) | 46.0% (40.2–51.9) | 26.7% (21.6–32.7) | Transient |
| | 4 | 1049 | 19.7% (16.4–23.5) | 7.2% (5.3–9.7) | 0.7% (0.3–1.6) | 1.4% (0.7–2.5) | Resolving |
| | 5 | 396 | 0.0% (0.0–0.9) | 0.0% (0.0–0.9) | 0.0% (0.0–0.9) | 0.7% (0.2–2.7) | Consistent |
| **Problem solving** | | | n = 1056 | n = 1182 | n = 1427 | n = 1364 | |
| | 1 | 72 | 75.5% (61.6–85.5) | 73.7% (60.8–83.5) | 87.9% (76.8–94.1) | 11.8% (5.4–23.8) | Resolving |
| | 2 | 11 | 100.0% (71.5–100.0) | 57.1% (23.0–85.6) | 90.0% (53.2–98.6) | 100.0% (71.5–100.0) | Consistent |
| | 3 | 39 | 100.0% (91.0–100.0) | 60.0% (41.9–75.7) | 0.0% (0.0–9.1) | 0.0% (0.0–9.1) | Resolving |
| | 4 | 1420 | 13.9% (11.6–16.5) | 11.4% (9.5–13.7) | 9.6% (8.0–11.6) | 1.6% (0.9–2.6) | Resolving |
| | 5 | 342 | 0.0% (0.0–1.1) | 0.0% (0.0–1.1) | 0.0% (0.0–1.1) | 0.0% (0.0–1.1) | Consistent |

n: Sample size, CI: Confidence interval.

[a]These proportions were used during description of the trajectory groups to gauge the transitions between risk levels. Emerging describes an increase in risk over time, resolving describes a decrease in risk over time, consistent describes relatively constant risk over time, and transient describes changing risk which may be both increasing or decreasing over time but is not consistent.

**Table 5. Risk ratios for the association of gestational age at birth (per week) with probability of group membership per individual ASQ-3 (ages & stages questionnaire 3rd edition) domain.**

| Domain | Trajectory group | crude RR (95% CI)[a] | adjusted RR (95% CI)[a,b] |
|---|---|---|---|
| Communication | 1: Emerging high risk | **1.49 (1.03–2.16)[c]** | 1.42 (0.95–2.10) |
| | 2: Transient high risk | **1.23 (1.02–1.48)[c]** | 1.18 (0.97–1.45) |
| | 3: Transient moderate risk | 1.05 (0.99–1.12) | 1.04 (0.98–1.12) |
| | 4: Consistent low risk | **REF** | **REF** |
| Personal social | 1: Resolving high risk | **1.21 (1.05–1.40)[c]** | 1.16 (1.00–1.35) |
| | 2: Transient moderate risk | **1.10 (1.02–1.17)[c]** | 1.08 (1.00–1.16) |
| | 3: Consistent high risk | 1.26 (0.87–1.82) | 1.16 (0.78–1.71) |
| | 4: Consistent low risk | **REF** | **REF** |
| Fine motor | 1: Emerging high risk | **1.54 (1.26–1.87)[c]** | **1.46 (1.19–1.80)[c]** |
| | 2: Transient high risk | 1.14 (0.99–1.31) | 1.13 (0.97–1.30) |
| | 3: Resolving moderate risk | **1.12 (1.03–1.21)[c]** | **1.11 (1.03–1.21)[c]** |
| | 4: Consistent low risk | **REF** | **REF** |
| Gross motor | 1: Resolving high risk | **1.17 (1.01–1.37)[c]** | **1.21 (1.04–1.42)[a]** |
| | 2: Consistent high risk | **1.65 (1.23–2.22)[c]** | **1.64 (1.20–2.24)[a]** |
| | 3: Transient high risk | 0.95 (0.85–1.07) | 0.95 (0.84–1.07) |
| | 4: Resolving moderate risk | 1.00 (0.91–1.09) | 0.99 (0.91–1.08) |
| | 5: Consistent low risk | **REF** | **REF** |
| Problem solving | 1: Resolving high risk A | **1.28 (1.06–1.54)[c]** | 1.18 (0.97–1.44) |
| | 2: Consistent high risk | **1.64 (1.14–2.34)[c]** | **1.58 (1.09–2.28)[c]** |
| | 3: Resolving high risk B | 1.07 (0.82–1.40) | 1.03 (0.79–1.35) |
| | 4: Resolving moderate risk | 1.10 (1.00–1.21) | 1.08 (0.98–1.19) |
| | 5: Consistent low risk | **REF** | **REF** |

RR: Risk ratio, CI: Confidence interval, REF: Referent group.

[b] The risk ratio can be interpreted as the rate of change in the probability of group membership per week before 40

[a] Risk ratios are adjusted for gestational age, maternal age, ethnicity, anxiety, depression, and infant sex.

weeks relative to the referent group.

[c] statistically significant at $\alpha = 0.05$.

what we might consider typical development, reaching those domain-specific developmental milestones at the appropriate chronological age. However, in all domains (except communication) most children within this study experienced changing levels of risk of developmental delay, which can be described as either resolving or transient. In a cross-sectional analysis, as with a one-time developmental screening program, these children might even appear typical. However, many of these children encounter domain-specific difficulties at one point before 5 years of age. The purpose of screening tools such as the ASQ-3 is to identify children with disabilities or difficulties and those at risk for further challenges. These at-risk children may not qualify for formal intervention programs, but that does not mean they do not require or would not benefit from intervention. However, further research on factors concerning these resolving and transient groups, aside from gestational age at birth, is necessary to determine who may benefit most and from which interventions.

## Strengths & limitations

Strengths of this study include the cohort study design, analyzing the continuous ASQ-3 scores, and using group-based trajectory modeling (GBTM) to estimate the association between gestational age and domain-specific developmental delay. The AOF cohort is a

Table 6. Distribution of sample characteristics by trajectory group when modeling individual ASQ-3 (ages & stages questionnaire 3rd edition) domains[a].

| Domain | Group | n | Child | | | | Maternal | | | | | | | Pregnancy | | | |
|---|---|---|---|---|---|---|---|---|---|---|---|---|---|---|---|---|---|
| | | | Gestational age Mean weeks (95% CI) | Sex % Male (95% CI) | SGA % Yes (95% CI) | NICU admission % Yes (95% CI) | Maternal age Mean weeks (95% CI) | Education % ≤ $40,000 (95% CI) | Household income % ≤ Highschool (95% CI) | Ethnicity % BIPOC (95% CI) | Parity % Yes (95% CI) | Anxiety % Yes (95% CI) | Depression % Yes (95% CI) | Stress % Yes (95% CI) | Binge drinking % Yes (95% CI) | Smoking % Yes (95% CI) | Drug use % Yes (95% CI) |
| **Communication** | 1: Emerging high risk | 25 | 37.9 (37.0–38.8) | 70.0 (37.6–90.0) | 11.1 (1.5–50.0) | 10.0 (1.4–46.7) | 31.2 (28.2–34.2) | 70.0 (37.6–90.0) | 70.0 (37.6–90.0)[b] | 60.0 (29.7–84.2)[b] | 50.0 (22.4–77.6) | 40.0 (15.8–70.3) | 30.0 (10.0–62.4)[b] | 30.0 (10.0–62.4) | 20.0 (5.0–54.1) | 30.0 (10.0–62.4) | 0.0 (0.0–13.7) |
| | 2: Transient high risk | 311 | 38.4 (38.0–38.8) | 75.9 (62.8–85.5)[b] | 13.5 (6.6–25.7) | 8.2 (3.1–19.8) | 30.6 (29.1–32.1) | 85.2 (73.1–92.4) | 94.2 (83.6–98.1) | 33.3 (22.1–46.8) | 43.4 (30.8–56.9) | 19.6 (10.9–32.8) | 15.1 (7.7–27.4) | 20.4 (11.7–33.2) | 9.3 (3.9–20.4) | 7.4 (2.8–18.1) | 1.9 (0.3–12.0) |
| | 3: Transient moderate risk | 522 | 38.7 (38.6–38.8) | 57.9 (54.6–61.2)[b] | 10.0 (8.1–12.3) | 4.6 (3.4–6.3) | 30.8 (30.5–31.1) | 90.7 (88.5–92.4) | 92.6 (90.7–94.2)[b] | 21.1 (18.5–23.9) | 45.8 (42.5–49.1) | 15.6 (13.3–18.2) | 6.3 (4.9–8.2)[b] | 15.6 (13.3–18.2) | 9.9 (8.1–12.1) | 12.0 (10.0–14.3) | 4.1 (2.9–5.6) |
| | 4: Consistent low risk | 1030 | 38.8 (38.7–38.8) | 48.4 (46.1–50.8)[b] | 11.2 (9.7–12.9) | 2.8 (2.1–3.8) | 30.7 (30.5–30.9) | 88.8 (87.2–90.2) | 91.3 (89.9–92.6) | 22.4 (20.5–24.5)[b] | 46.8 (44.4–49.2) | 15.4 (13.7–17.2) | 8.0 (6.8–9.3)[b] | 17.3 (15.6–19.2) | 10.2 (8.8–11.9) | 11.5 (10.0–13.3) | 3.9 (3.0–4.9) |
| **Personal social** | 1: Resolving high risk | 114 | 38.5 (38.2–38.8) | 59.6 (50.4–68.2) | 12.0 (7.1–19.6) | 7.6 (3.9–14.5)[b] | 31.5 (30.6–32.3) | 89.4 (82.2–93.9) | 93.6 (87.2–96.9) | 23.7 (16.8–32.3) | 35.1 (26.9–44.3)[b] | 20.9 (14.3–29.5) | 15.0 (9.6–22.9)[b] | 21.9 (15.3–30.5)[b] | 6.2 (3.0–12.4) | 8.8 (4.8–15.7) | 0.9 (0.1–6.0) |
| | 2: Transient moderate risk | 967 | 38.7 (38.6–38.7)[b] | 56.9 (54.5–59.2) | 11.6 (10.1–13.3) | 3.9 (3.0–4.9) | 30.7 (30.4–30.9) | 88.8 (87.2–90.2) | 90.1 (88.6–91.5)[b] | 24.4 (22.5–26.5)[b] | 46.9 (44.5–49.3)[b] | 16.8 (15.0–18.7)[b] | 8.2 (7.0–9.6)[b, c] | 18.4 (16.6–20.3)[b] | 10.1 (8.7–11.8) | 12.9 (11.3–14.7) | 4.1 (3.2–5.2) |
| | 3: Consistent high risk | 14 | 38.4 (37.7–39.2) | 78.6 (50.6–92.9) | 7.1 (1.0–37.1) | 7.1 (1.0–37.1) | 32.7 (30.4–35.0) | 78.6 (50.6–92.9) | 92.9 (62.9–99.0) | 42.9 (20.6–68.4)[b] | 35.7 (15.7–62.4) | 50.0 (26.0–74.0)[b] | 35.7 (15.7–62.4)[c] | 38.5 (17.0–65.7)[b] | 21.4 (7.1–49.4) | 21.4 (7.1–49.4) | 0.0 (0.0–23.2) |
| | 4: Consistent low risk | 793 | 38.8 (38.7–38.9)[b] | 40.5 (37.1–43.9)[b] | 8.9 (7.0–11.2)[b] | 2.3 (1.4–3.7)[b] | 30.8 (30.5–31.1) | 90.6 (88.4–92.5) | 95.0 (93.2–96.3)[b] | 17.3 (14.8–20.1)[b] | 47.2 (43.7–50.7) | 11.8 (9.7–14.3)[b] | 4.9 (3.6–6.7)[b, c] | 12.5 (10.4–15.0)[b] | 10.5 (8.5–12.9) | 9.6 (7.7–11.9) | 4.0 (2.8–5.6) |
| **Fine motor** | 1: Emerging high risk | 44 | 38.0 (37.6–38.5)[b] | 84.1 (70.2–92.2)[b] | 9.3 (3.5–22.3) | 17.1 (8.4–31.7)[b] | 30.3 (29.1–31.4) | 81.8 (67.6–90.6) | 93.0 (80.5–97.7) | 15.9 (7.8–29.8) | 59.1 (44.2–72.5) | 25.6 (14.8–40.6) | 15.9 (7.8–29.8) | 29.5 (18.0–44.5)[c] | 13.6 (6.3–27.2) | 25.0 (14.4–39.8)[b] | 4.5 (1.1–16.4) |
| | 2: Transient high risk | 162 | 38.7 (38.5–38.9)[b] | 72.2 (64.8–78.6)[c] | 14.6 (9.9–21.0) | 6.3 (3.4–11.2) | 30.9 (30.2–31.6) | 89.4 (83.6–93.3) | 90.1 (84.3–94.0) | 24.7 (18.7–31.9) | 51.6 (43.8–59.2) | 19.6 (14.1–26.6) | 9.9 (6.1–15.5) | 24.8 (18.8–32.1)[c] | 10.5 (6.6–16.2) | 11.7 (7.6–17.7) | 4.3 (2.1–8.8) |
| | 3: Resolving moderate risk | 1138 | 38.7 (38.6–38.8)[b] | 52.8 (50.5–55.0)[b, c] | 11.0 (9.6–12.6) | 3.2 (2.4–4.1)[b] | 30.6 (30.4–30.8)[b] | 88.5 (87.0–89.8)[b] | 90.9 (89.5–92.1)[b] | 23.5 (21.7–25.5)[b] | 45.2 (42.9–47.5) | 15.5 (13.9–17.3) | 7.8 (6.7–9.1) | 16.7 (15.1–18.5)[b] | 10.4 (9.0–12.0) | 12.0 (10.5–13.6)[b] | 3.8 (3.0–4.7) |
| | 4: Consistent low risk | 539 | 38.9 (38.8–39.0)[b] | 41.6 (37.5–45.8)[b, c] | 9.5 (7.2–12.4) | 3.0 (1.8–4.9)[b] | 31.3 (31.0–31.7)[b] | 92.7 (90.2–94.6)[b] | 95.0 (92.7–96.6)[b] | 18.0 (15.0–21.5)[b] | 48.1 (43.9–52.4) | 14.0 (11.2–17.2) | 5.8 (4.1–8.1) | 13.9 (11.2–17.2)[c] | 8.8 (6.7–11.6) | 9.6 (7.4–12.4)[b] | 4.1 (2.7–6.2) |

(Continued)

**Table 6.** (Continued)

| Domain | Group | n | Child | | | | Maternal | | | | | | | Pregnancy | | | |
|---|---|---|---|---|---|---|---|---|---|---|---|---|---|---|---|---|---|
| | | | Gestational age | Sex | SGA | NICU admission | Maternal age | Education | Household income | Ethnicity | Parity | Anxiety | Depression | Stress | Binge drinking | Smoking | Drug use |
| | | | Mean weeks (95% CI) | % Male (95% CI) | % Yes (95% CI) | % Yes (95% CI) | Mean weeks (95% CI) | % ≤ $40,000 (95% CI) | % ≤ Highschool (95% CI) | % BIPOC (95% CI) | % Yes (95% CI) | % Yes (95% CI) | % Yes (95% CI) | % Yes (95% CI) | % Yes (95% CI) | % Yes (95% CI) | % Yes (95% CI) |
| Gross motor | 1: Resolving high risk | 109 | **38.4 (38.2–38.7)**[b] | **42.2 (33.3–51.6)**[b] | 11.9 (6.9–19.8) | 3.9 (1.5–9.9) | **31.9 (31.0–32.8)**[b] | 89.9 (82.7–94.3) | 96.3 (90.5–98.6) | **11.9 (7.1–19.5)**[b] | **34.9 (26.5–44.3)**[b] | 11.2 (6.5–18.7)[b] | 6.4 (3.1–12.9)[b] | 10.1 (5.7–17.3) | 9.2 (5.0–16.2) | 8.3 (4.4–15.1) | 1.9 (0.5–7.1) |
| | 2: Consistent high risk | 15 | **37.5 (36.7–38.4)**[b] | 60.0 (34.8–80.8) | 14.3 (3.6–42.7) | 13.3 (3.4–40.6) | 30.7 (28.5–32.9) | 100.0 (78.2–100.0) | 86.7 (59.4–96.6) | 46.7 (24.1–70.7)[b, c] | 53.3 (29.3–75.9) | 50.0 (26.0–74.0)[b] | 33.3 (14.6–59.4)[b] | 26.7 (10.4–53.3) | 6.7 (0.9–35.2) | 6.7 (0.9–35.2) | 0.0 (0.0–21.8) |
| | 3: Consistent moderate risk | 318 | 38.8 (38.7–38.9)[b, c] | 58.2 (52.7–63.5)[b] | 14.4 (10.9–18.9) | 4.1 (2.3–7.1) | 30.7 (30.2–31.2) | 89.2 (85.3–92.2) | 92.8 (89.3–95.2) | 19.8 (15.8–24.6) | 51.1 (45.6–56.6)[b] | 15.1 (11.5–19.5)[b] | 8.9 (6.2–12.6)[b] | 16.8 (13.0–21.4) | 11.1 (8.1–15.1) | 11.8 (8.7–15.8) | 3.5 (1.9–6.2) |
| | 4: Resolving moderate risk | 1049 | 38.7 (38.7–38.8)[b, c] | 51.9 (49.6–54.2) | 10.3 (8.9–11.9) | 3.1 (2.4–4.1) | 30.6 (30.4–30.8)[b] | 89.0 (87.5–90.4) | 91.2 (89.7–92.4) | 24.1 (22.2–26.1)[b] | 46.3 (44.0–48.6) | 16.1 (14.4–17.9)[b] | 7.6 (6.5–9.0)[b] | 17.9 (16.2–19.7) | 9.6 (8.2–11.1) | 11.7 (10.1–13.4) | 4.0 (3.1–5.0) |
| | 5: Consistent low risk | 396 | 38.7 (38.6–38.8)[b, c] | 51.0 (46.1–55.9) | 9.9 (7.3–13.5) | 4.6 (2.9–7.2) | 31.1 (30.6–31.5) | 89.8 (86.5–92.5) | 92.4 (89.2–94.6) | **18.4 (14.9–22.6)**[c] | 46.3 (41.4–51.3) | 14.0 (10.9–17.9)[b] | 6.1 (4.1–9.0)[b] | 13.8 (10.8–17.6) | 11.9 (9.1–15.5) | 12.7 (9.7–16.4) | 4.6 (2.9–7.1) |
| Problem solving | 1: Resolving high risk A | 72 | 38.4 (38.1–38.8) | **66.7 (55.1–76.6)**[b] | 8.8 (4.0–18.3) | 6.0 (2.3–14.9) | 31.1 (29.9–32.2) | 87.5 (77.7–93.4) | 85.5 (75.1–92.0) | 33.3 (23.4–44.9)[b] | 54.2 (42.6–65.3) | 20.0 (12.2–31.0) | 15.3 (8.7–25.5) | 18.1 (10.8–28.7) | 12.5 (6.6–22.3) | 9.7 (4.7–19.0) | 1.4 (0.2–9.2) |
| | 2: Consistent high risk | 11 | **37.8 (36.9–38.7)**[b] | 81.8 (49.3–95.4) | 20.0 (5.0–54.1) | 9.1 (1.3–43.9) | 30.2 (27.9–32.4) | 81.8 (49.3–95.4) | 81.8 (49.3–95.4) | 27.3 (9.0–58.6) | 27.3 (9.0–58.6) | 27.3 (9.0–58.6) | 18.2 (4.6–50.7) | 18.2 (4.6–50.7) | 27.3 (9.0–58.6) | 18.2 (4.6–50.7) | 0.0 (0.0–28.5) |
| | 3: Resolving high risk B | 39 | 38.7 (38.4–39.1)[b] | 56.4 (40.7–70.9) | 8.1 (2.6–22.3) | 2.6 (0.4–16.5) | 31.9 (30.3–33.5) | 94.7 (81.2–98.7) | 92.3 (78.7–97.5) | 38.5 (24.7–54.4)[b] | 38.5 (24.7–54.4) | 20.5 (10.6–36.0) | 7.7 (2.5–21.3) | 23.1 (12.5–38.7) | 2.6 (0.4–16.5) | 7.9 (2.6–21.8) | 2.6 (0.4–16.5) |
| | 4: Resolving moderate risk | 1420 | 38.7 (38.7–38.8)[b] | 52.1 (50.0–54.2)[b] | 11.1 (9.7–12.5) | 3.8 (3.0–4.7) | 30.7 (30.5–30.8) | 88.9 (87.6–90.2) | **91.4 (90.1–92.5)**[b] | 23.0 (21.3–24.8)[b] | 46.4 (44.3–48.5) | 16.0 (14.5–17.6) | 7.6 (6.6–8.8) | **17.7 (16.1–19.3)**[b] | 10.2 (8.9–11.6) | **12.5 (11.1–14.0)**[b] | 4.1 (3.3–5.1) |
| | 5: Consistent low risk | 342 | 38.8 (38.7–39.0)[b] | 48.2 (43.0–53.5)[b] | 10.3 (7.5–14.1) | 1.9 (0.8–4.1) | 31.1 (30.7–31.6) | 91.4 (88.0–94.0) | 95.7 (92.8–97.4)[b] | **14.0 (10.7–18.1)**[b] | 46.4 (41.2–51.8) | 11.5 (8.5–15.4) | 5.9 (3.8–9.0) | 10.7 (7.9–14.5)[b] | 9.7 (7.0–13.3) | 7.6 (5.3–11.0)[b] | 3.2 (1.8–5.7) |

SGA: Small for gestational age, NICU: Neonatal intensive care, BIPOC: Black, Indigenous, People of Colour, CI: Confidence interval.

[a] This table was used to guide initial covariate selection when modeling the association between gestational age at birth and the probability of group membership within domain specific trajectory groups.

[b] Bolded values are considered significantly different from the non-bolded values marked with the corresponding superscript as determined by the exclusivity of the 95% confidence intervals.

[c] Bolded values are considered significantly different from the non-bolded values marked with the corresponding symbol as determined by the exclusivity of the 95% confidence intervals.

longitudinal prospective pregnancy cohort that followed all 2664 mother-child dyads included in this study from pregnancy until their child was 5 years old, consistently measuring the risk of developmental delay with the ASQ-3 at four separate time points [16]. Utilizing prospective data collection minimized temporal bias of confounding factors, gestational age at birth, and developmental delay outcomes [34].

The ASQ-3 scores within each domain range from 0 to 60, and most of the research using this tool to measure developmental delays in children use a cut-off of 2 standard deviations from the normalized mean to indicate increased risk for developmental delay [1,2,14,15]. The purpose of a screening tool, such as the ASQ-3 is to capture all persons possibly at risk [35]. A 2018 study found that using 1 SD rather than 2 SD enhanced the test characteristics as a screening tool for neurodevelopmental disabilities as well as developmental delay [36].

Trajectories of risk of developmental delay were modeled in the current paper using the 0 to 60 domain-specific scores and were described in relation to the normalized mean cut-offs using the 1 SD cut-off to convey the level of risk of developmental delay. In using the continuous rather than the dichotomized scores, this study determined that increased risk of delay within a domain is rarely consistent, and using continuous data provides a more nuanced understanding of the risk of developmental delay.

GBTM compliments the cohort study design in this analysis. The longitudinal data enabled the application of GBTM to analyze the outcome of interest over four time points; thus, enabling cubic polynomial orders during modeling for greater precision in the group membership probabilities of trajectory groups [28]. Further, GBTM reduces the effect of attrition or non-participation bias inherent to longitudinal cohorts as it accommodates missing data due to intermittent or missed assessments [37]. Using GBTM, rather than fixed/random/mixed-effects longitudinal regression models, marginal models, or traditional growth curve models for analysis, we identified resolving, emerging, and transient patterns of increased risk which would otherwise be missed by methods requiring a priori grouping [38].

Certain limitations have been identified within this study's design, including analytical power, use of the ASQ-3 to measure developmental delay, and generalization of the study's findings. As we cannot a priori determine sample sizes for trajectory groups using GBTM, there are relatively small sample sizes within some trajectory groups. While these smaller trajectory group sizes within the high emerging or high transient groups within the present study are aligned with those within prior research on stability of risk of developmental delay[32], this particular analytical method has not been employed in a larger cohort to our knowledge so we cannot discount the possibility that within a larger cohort alternative trajectory groups may be observed. Additionally, trajectory group sample may have contributed to a lack of sufficient power within regression models to determine a significant association between gestational age and trajectory group membership when adjusting for confounding factors. The sensitivity of the ASQ-3 can change with age of assessment [39–41]; with previous research identifying increased sensitivity of the screening tool at age three, and as the sensitivity increases, there will be more children identified as "at risk of delay" [42]. Though this potentially could explain the increased risk at age 3 seen in the "transient high risk" trajectory group within the fine motor domain in this study, the limitations of psychometric consistency throughout each domain over time should be noted. And finally, though at recruitment participants within the AOF cohort represented the demographic characteristics of urban Alberta, Canada, the elevated median socioeconomic status of this cohort's population may limit the generalizability of this study's findings [16]. As prior research has shown that lower socioeconomic status increases the risk of early childhood developmental delay [43], the cohort's elevated socioeconomic status may underestimate the association between gestational age and risk of developmental delay.

## Conclusions

Development is a longitudinal, multifaceted process. To evaluate a potential risk factor for delayed development, such as birth before FT, we need to account for the longitudinal nature of development. In this study, the risk of developmental delay was modeled as trajectories unique to each of five separate domains of development. Most children experienced changing levels of risk for delay over time, as opposed to having a consistent low risk pattern. These same children would appear as though they are meeting development milestones if cross-sectional analyses were used along with a dichotomized view of developmental delay. However, our findings highlight that many of these children may encounter domain-specific difficulties at some point by 5 years of age. Although birth before FT is a threat to health and development, the domains most likely to experience delay are related to fine and gross motor development and problem solving. The anticipated trajectories for children born ≥34 and <41 weeks of gestation should also consider several demographic and lifestyle factors.

## Acknowledgments

The authors gratefully acknowledge the All Our Families study team as well as the cohort's participants and their families.

## Author Contributions

**Conceptualization:** Nikki L. Stephenson, Suzanne Tough, Amy Metcalfe.

**Data curation:** Nikki L. Stephenson, Suzanne Tough, Sheila McDonald.

**Formal analysis:** Nikki L. Stephenson.

**Funding acquisition:** Nikki L. Stephenson, Suzanne Tough.

**Investigation:** Nikki L. Stephenson.

**Methodology:** Nikki L. Stephenson, Tyler Williamson, Sheila McDonald, Carly McMorrris, Amy Metcalfe.

**Project administration:** Nikki L. Stephenson.

**Resources:** Suzanne Tough.

**Supervision:** Suzanne Tough, Amy Metcalfe.

**Visualization:** Nikki L. Stephenson.

**Writing – original draft:** Nikki L. Stephenson.

**Writing – review & editing:** Nikki L. Stephenson, Suzanne Tough, Tyler Williamson, Sheila McDonald, Carly McMorrris, Amy Metcalfe.

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
