## [Decision Letter · Decision Letter 0]

20 Jul 2023

PONE-D-23-18682Early Childhood Trajectories of Domain-Specific Developmental Delay and Gestational Age at Birth: An Analysis of the All Our Families CohortPLOS ONE

Dear Dr. Stephenson,

Thank you for submitting your manuscript to PLOS ONE. After careful consideration, we feel that it has merit but does not fully meet PLOS ONE’s publication criteria as it currently stands. Therefore, we invite you to submit a revised version of the manuscript that addresses the points raised during the review process.

ACADEMIC EDITOR: It is an interesting study where authors investigated the developmental trajectories in children according to gestational age at birth. However, some technical issues have been raised by the reviewers. Authors are suggested to address them carefully, and improve the manuscript accordingly.

We look forward to receiving your revised manuscript.

Kind regards,

Shaonong Dang, PhD

Academic Editor

PLOS ONE

2. Please amend either the title on the online submission form (via Edit Submission) or the title in the manuscript so that they are identical.

Additional Editor Comments:

It is an interesting study where authors investigated the developmental trajectories in children according to gestational age at birth. However, some technical issues have been raised by the reviewers. Authors are suggested to address them carefully, and improve the manuscript accordingly.

Reviewers' comments:

Reviewer's Responses to Questions

**Comments to the Author**

1. Is the manuscript technically sound, and do the data support the conclusions?

Reviewer #1: Yes

Reviewer #2: Yes

Reviewer #3: Partly

2. Has the statistical analysis been performed appropriately and rigorously? 

Reviewer #1: Yes

Reviewer #2: I Don't Know

Reviewer #3: Yes

3. Have the authors made all data underlying the findings in their manuscript fully available?

Reviewer #1: No

Reviewer #2: No

Reviewer #3: Yes

4. Is the manuscript presented in an intelligible fashion and written in standard English?

Reviewer #1: Yes

Reviewer #2: Yes

Reviewer #3: Yes

5. Review Comments to the Author

Reviewer #1: The authors have provided a well-conducted analysis of longitudinal data exploring gestational age and development. The analysis is conducted across several developmental domains: communication, problem solving, personal-social, fine motor, and gross motor. The work helps contribute to a growing understanding of the both the risks posed by late preterm and early preterm birth as well as the vulnerability to delay that can occur across early childhood. The findings also offer an indication of where the focus in domain specific intervention might take.

Reviewer #2: Thank you for the opportunity to review this interesting work, looking at developmental trajectories in children according to gestational age at birth, using the different domains assessed in the Ages and Stages questionnaire at points between 1 and 5 years of age. This is a very important area of work, given the mounting evidence that outcomes in early childhood are inversely proportional to gestational age. It is now clear that gestational age is a continuum, rather than a dichotomy, and so it is increasingly important to know more detail about the types of problems encountered by these children, and what happens to them as they get older, in order to determine whether, how and when to target interventions to support their health, educational and developmental needs.

I am not a statistician or developmental psychologist, and am not familiar with group-based trajectory modeling as a method of analysis. I will therefore limit this review to comments on the clinical aspects and implications of the work.

Introduction

The authors appropriately summarise the background literature regarding outcomes according to gestational age, and in particular the now recognised effects associated with late preterm birth, as well as limitations in this knowledge.

Results

The large tables and limited explanatory text are somewhat challenging for readers who are unfamiliar with the methodology used. Could they perhaps be simplified, or explained more fully in the text? Figure 2 provides a more accessible diagramatic representation that is helpful to understanding of the complexities of the results. Overall, the results appear to be consistent with other work in the field and show a relationship between gestational age and outcomes. This paper may go some way to clarifying the particular issues faced by children born late preterm and determining where, within the wide spectrum of late preterm birth, the main issues lie.

It is interesting to see that, in this cohort, risks appear to change with age and that many resolve, but also that a substantial number encounter some domain-specific problems at some point before they reach 5 years of age.

Discussion

The authors have appropriately highlighted some limitations to their work. The limitations of most concern is the relatively small numbers falling within some of the groups, which invites the question of whether similar results would be seen in a much larger cohort, or whether this might demonstrate more consistency in the children's developmental trajectories. The authors state that the area studied has a relatively affluent population, and rightly suggest that this may affect the reulsts, since it is well-accepted that trajectories may be modified considerably depending on the socioeconomic status of the family. These results may therefore not be applicable to a less well-off or educated population.

General comments

The paper is well written and raises some interesting points. While there are limitations, the results begin to unpick the complex nature of adverse outcomes in more mature preterm infants, and may generate hypotheses that pave the way for future work on this subject.

Reviewer #3: Thank you for asking me to review this manuscript which discusses an interesting topic. The paper is well-written. However, I have some issues with the conceptualisation of the analysis and therefore the practical relevance to practice. I would have thought that based on the ‘introduction’ the authors would pursue the analysis in the line of demonstrating how risks / trajectories change based on late preterm, PT (≥34 to < 37) and early term, ET (≥37 to <39 weeks) referent to full-term (defined as ≥39 to <41). Surprising, this was not done. Merely indicating that the trajectories of developmental delays with decreasing gestation age doesn’t add much value to knowledge. Can we categorise the GA as exposure? Is it possible to know if late PT and early term babies have differing risks / trajectories to FT, and can authors also directly compare the risks between PT and ET? I think that would be more helpful for practice.

6. PLOS authors have the option to publish the peer review history of their article (what does this mean?). If published, this will include your full peer review and any attached files.

Reviewer #1: No

Reviewer #2: No

Reviewer #3: No

---

## [Author Response · Author response to Decision Letter 0]

18 Sep 2023

JOURNAL REQUIREMENTS:

Author response: We have read the PLOS ONE's style requirements, and have revised the manuscript to ensure this manuscript meets said requirements.

2. Please amend either the title on the online submission form (via Edit Submission) or the title in the manuscript so that they are identical.

Author response: We have amended the title on the submission form and manuscript to ensure they are identical.

ADDITIONAL EDITOR COMMENTS:

1. It is an interesting study where authors investigated the developmental trajectories in children according to gestational age at birth. However, some technical issues have been raised by the reviewers. Authors are suggested to address them carefully, and improve the manuscript accordingly.

Author response: Thank you for the opportunity to improve this manuscript through revision and resubmit to PLOS ONE, we address issues raised by reviewers below.

REVIEWER'S RESPONSES TO QUESTIONS

1. Is the manuscript technically sound, and do the data support the conclusions?

Reviewer #1: Yes

Reviewer #2: Yes

Reviewer #3: Partly

2. Has the statistical analysis been performed appropriately and rigorously?

Reviewer #1: Yes

Reviewer #2: I Don't Know

Reviewer #3: Yes

3. Have the authors made all data underlying the findings in their manuscript fully available?

Reviewer #1: No

Reviewer #2: No

Reviewer #3: Yes

Author response: Reviewers #1 and #2 noted that all data underlying the findings in the present manuscript was not made fully available. As we note within the data availability statement, data cannot be shared publicly because of legal and ethical agreements. 

Data that support the findings of this study are available upon request from the All Our Families Cohort (https://allourfamiliesstudy.com/data-access/). Requests must undergo ethical review before entering into a data-sharing agreement with the institution to protect the privacy and confidentiality of the cohort’s participants. The metadata of variables used within this study can be found at https://www.maelstrom-research.org/study/aob. 

As per PLOS ONE’s “Acceptable Data Access Restrictions”, we have described these restrictions within the “Data availability statement” as well as the “Participant consent and ethics statement” section of the manuscript (lines 107-114 within the original or 112-118 after revisions). We hope that Reviewers #1 and #2 are amenable to this point on data sharing, as we are unable to make the underlying data set publicly available for both legal and ethical reasons.

4. Is the manuscript presented in an intelligible fashion and written in standard English?

Reviewer #1: Yes

Reviewer #2: Yes

Reviewer #3: Yes

REVIEWER COMMENTS TO THE AUTHOR:

Reviewer #1: 

1. The authors have provided a well-conducted analysis of longitudinal data exploring gestational age and development. The analysis is conducted across several developmental domains: communication, problem solving, personal-social, fine motor, and gross motor. The work helps contribute to a growing understanding of the both the risks posed by late preterm and early preterm birth as well as the vulnerability to delay that can occur across early childhood. The findings also offer an indication of where the focus in domain specific intervention might take.

Author response: We thank Reviewer #1 for their comments. 

Reviewer #2: 

1. Thank you for the opportunity to review this interesting work, looking at developmental trajectories in children according to gestational age at birth, using the different domains assessed in the Ages and Stages questionnaire at points between 1 and 5 years of age. This is a very important area of work, given the mounting evidence that outcomes in early childhood are inversely proportional to gestational age. It is now clear that gestational age is a continuum, rather than a dichotomy, and so it is increasingly important to know more detail about the types of problems encountered by these children, and what happens to them as they get older, in order to determine whether, how and when to target interventions to support their health, educational and developmental needs.

Author response: We would like to thank Reviewer #2 for this comment.

2. The large tables and limited explanatory text are somewhat challenging for readers who are unfamiliar with the methodology used. Could they perhaps be simplified, or explained more fully in the text? Figure 2 provides a more accessible diagramatic representation that is helpful to understanding of the complexities of the results. Overall, the results appear to be consistent with other work in the field and show a relationship between gestational age and outcomes. This paper may go some way to clarifying the particular issues faced by children born late preterm and determining where, within the wide spectrum of late preterm birth, the main issues lie.

Author response: We added additional text in the Results section under “Description of trajectory groups by model” to explain the tables. 

3. The authors have appropriately highlighted some limitations to their work. The limitations of most concern is the relatively small numbers falling within some of the groups, which invites the question of whether similar results would be seen in a much larger cohort, or whether this might demonstrate more consistency in the children's developmental trajectories. The authors state that the area studied has a relatively affluent population, and rightly suggest that this may affect the results, since it is well-accepted that trajectories may be modified considerably depending on the socioeconomic status of the family. These results may therefore not be applicable to a less well-off or educated population.

Author response: We would like to thank Reviewer #2 for their thoughtful comments. As Reviewer #2 commented, the relatively small numbers within some of the groups invite the question of whether similar results would be seen in a larger cohort or whether this (the small sample size within the trajectory group) might demonstrate more consistency in the children's developmental trajectories. During model specification, we specified the base number of groups using the Bayesian Information Criterion and the Akaike Information Criterion values, but we also visually inspected these groups by graphing the mean values and confidence intervals throughout the model selection process to ensure that we were not separating or combining distinct groups based solely on those criterion values. If repeating this process in a larger cohort we may see similar results if applying this same method, as the sample sizes within the high emerging or high transient groups within the present study are similar to that seen within the persistent and emerging groups reported by Hornman et al in 2017 where they investigated the stability of ASQ-3 scores at ages 4 and 5 within the Longitudinal Preterm Outcome Project (LOLLIPOP) cohort study. However, as we cannot increase the cohort sample size we cannot say for certain. Therefore, we have expanded the methods section to provide more information on model selection, and the limitations section to include more discussion on the role of sample size.

In response to the second comment, “These results may therefore not be applicable to a less well-off or educated population,” we have expanded the limitations section on how this affluence biases the presented results. 

Reviewer #3: 

1. Thank you for asking me to review this manuscript which discusses an interesting topic. The paper is well-written. However, I have some issues with the conceptualisation of the analysis and therefore the practical relevance to practice. I would have thought that based on the ‘introduction’ the authors would pursue the analysis in the line of demonstrating how risks / trajectories change based on late preterm, PT (≥34 to < 37) and early term, ET (≥37 to <39 weeks) referent to full-term (defined as ≥39 to <41). Surprising, this was not done. Merely indicating that the trajectories of developmental delays with decreasing gestation age doesn’t add much value to knowledge. Can we categorise the GA as exposure? Is it possible to know if late PT and early term babies have differing risks / trajectories to FT, and can authors also directly compare the risks between PT and ET? I think that would be more helpful for practice.

Author response: Thank you so much for your comment. We agree that the late preterm and early term infants are not the same as full-term born infants, however even within these broad categories our study shows that there are important differences. We feel that by analysing gestational age as a continuous variable in weeks is potentially more relevant for clinical practice, as it allows clinicians to provide targeted information for an individualized care plan for their patients using specific gestational ages.

---

## [Decision Letter · Decision Letter 1]

16 Oct 2023

PONE-D-23-18682R1Early childhood trajectories of domain-specific developmental delay and gestational age at birth: An analysis of the All Our Families cohortPLOS ONE

Dear Dr. Stephenson,

Thank you for submitting your manuscript to PLOS ONE. After careful consideration, we feel that it has merit but does not fully meet PLOS ONE’s publication criteria as it currently stands. Therefore, we invite you to submit a revised version of the manuscript that addresses the points raised during the review process.

Authors should address the comments further.

We look forward to receiving your revised manuscript.

Kind regards,

Shaonong Dang, PhD

Academic Editor

PLOS ONE

Additional Editor Comments:

Athough authors have addressed the comments from the reveiwers, some new concerns have been raised by the reviewer. Authors should further address them carefully to improve the manuscript.

Reviewers' comments:

Reviewer's Responses to Questions

**Comments to the Author**

1. If the authors have adequately addressed your comments raised in a previous round of review and you feel that this manuscript is now acceptable for publication, you may indicate that here to bypass the “Comments to the Author” section, enter your conflict of interest statement in the “Confidential to Editor” section, and submit your "Accept" recommendation.

Reviewer #2: All comments have been addressed

Reviewer #3: (No Response)

2. Is the manuscript technically sound, and do the data support the conclusions?

Reviewer #2: Yes

Reviewer #3: Partly

3. Has the statistical analysis been performed appropriately and rigorously? 

Reviewer #2: I Don't Know

Reviewer #3: Yes

4. Have the authors made all data underlying the findings in their manuscript fully available?

Reviewer #2: No

Reviewer #3: (No Response)

5. Is the manuscript presented in an intelligible fashion and written in standard English?

Reviewer #2: Yes

Reviewer #3: Yes

6. Review Comments to the Author

Reviewer #2: (No Response)

Reviewer #3: It appears there is a fundamental issue with the sample size, hence the reluctance of authors not to examine how the specific timings of birth – late preterm, PT (≥34 to < 37), and early term, ET (≥37 to <39 weeks), referent to full-term (defined as ≥39 to <41) relates with developmental trajectory. More so, that was their line of argument in the introduction. If the categorisation is irrelevant to clinical practice, then it shouldn’t be in the introduction. In addition, the third point raised by Review #3 was not properly addressed in the limitation section – generalisability issues. I think we should be clear if the current findings are generalisable.

This is a well-designed study, however, there seems to be hidden fundamental issue with the sample size that brings into question the generalisability of the findings and relevance to practice. This manuscript is likely to generate a lot of queries with readers.

7. PLOS authors have the option to publish the peer review history of their article (what does this mean?). If published, this will include your full peer review and any attached files.

Reviewer #2: No

Reviewer #3: No

---

## [Author Response · Author response to Decision Letter 1]

25 Oct 2023

Reviewer #3: It appears there is a fundamental issue with the sample size, hence the reluctance of authors not to examine how the specific timings of birth – late preterm, PT (≥34 to < 37), and early term, ET (≥37 to <39 weeks), referent to full-term (defined as ≥39 to <41) relates with developmental trajectory. More so, that was their line of argument in the introduction. If the categorisation is irrelevant to clinical practice, then it shouldn’t be in the introduction. In addition, the third point raised by Review #3 was not properly addressed in the limitation section – generalisability issues. I think we should be clear if the current findings are generalisable.

This is a well-designed study, however, there seems to be hidden fundamental issue with the sample size that brings into question the generalisability of the findings and relevance to practice. This manuscript is likely to generate a lot of queries with readers.

Response:

Thank you for the comments. We respectfully disagree that this is an issue of sample size; the sample sizes according to gestational age categories are displayed in Table 1. Table 1 is stratified by gestational age categories to inform readers of the distribution of the sample characteristics across the range of gestational ages, and the sample sizes within each of the gestational age categories provide readers with additional information on the distribution of the exposure than if only reporting a mean � SD. The planned analysis chose to include gestational age as a measured/continuous variable a priori for the following reasons:

1) Gestational age is a biological continuum, and while we acknowledge that categories are used for clinical decision-making, these categories are somewhat arbitrary as an infant born at 36 6/7 weeks of gestation compared to an infant born at 37 0/7 are categorized into two separate groups and are clinically treated differently, though their births may only be minutes apart.

2) Apgar scores, a routinely used measure of health status of newborns, are divided into clinically accepted and widely used categories; similar to gestational age. However, studies show increased risks of neonatal morbidity and mortality with lower Apgar scores among infants within the same Apgar category.[1-4] The present analysis, using gestational age by week, speaks to similar issues with using gestational age as a categorical variable. 

3) This paper provides new information, where many studies assess differences between gestational age categories, these results show that the risk of developmental delay increases incrementally with each week of gestational age. A literature review identified 8 systematic reviews for LPT births, 2 for ET terms, and 2 more for LPT and ET combined.[5-16] The authors of those reviews state that drawing conclusions from the aggregated data was difficult due to limitations such as lack of uniformity in reporting gestational age categories.[15] The present study's findings add to the body of evidence by reporting results per week of gestational age to reduce those identified limitations. 

While we had referred to gestational age categories within the introduction to describe the current body of evidence (which predominantly reports estimates by gestational age categories), we have made minor changes to the introduction and methods to reflect our a priori planned analysis using gestational age as a continuous variable.

With regards to the Reviewer’s comment, “the third point raised by Review #3 was not properly addressed in the limitation section”, we are unclear as to which comment the Reviewer is referring to as Reviewer #3 only had one comment in the last revision. If you could provide clarification, we will respond accordingly.

References

1. Persson M, Razaz N, Tedroff K, Joseph KS, Cnattingius S. Five and 10 minute Apgar scores and risks of cerebral palsy and epilepsy: population based cohort study in Sweden. Bmj. 2018;360:k207. Epub 20180207. doi: 10.1136/bmj.k207. PubMed PMID: 29437691; PubMed Central PMCID: PMCPMC5802319.

2. Razaz N, Boyce WT, Brownell M, Jutte D, Tremlett H, Marrie RA, et al. Five-minute Apgar score as a marker for developmental vulnerability at 5 years of age. Arch Dis Child Fetal Neonatal Ed. 2016;101(2):F114-20. Epub 20150717. doi: 10.1136/archdischild-2015-308458. PubMed PMID: 26187935; PubMed Central PMCID: PMCPMC4789716.

3. Razaz N, Cnattingius S, Joseph KS. Association between Apgar scores of 7 to 9 and neonatal mortality and morbidity: population based cohort study of term infants in Sweden. Bmj. 2019;365:l1656. Epub 20190507. doi: 10.1136/bmj.l1656. PubMed PMID: 31064770; PubMed Central PMCID: PMCPMC6503461.

4. Razaz N, Cnattingius S, Persson M, Tedroff K, Lisonkova S, Joseph KS. One-minute and five-minute Apgar scores and child developmental health at 5 years of age: a population-based cohort study in British Columbia, Canada. BMJ Open. 2019;9(5):e027655. Epub 20190509. doi: 10.1136/bmjopen-2018-027655. PubMed PMID: 31072859; PubMed Central PMCID: PMCPMC6528022.

5. Boswinkel V, Nijboer-Oosterveld J, Nijholt IM, Edens MA, Mulder-de Tollenaer SM, Boomsma MF, et al. A systematic review on brain injury and altered brain development in moderate-late preterm infants. Early Hum Dev. 2020;148:105094. Epub 2020/07/28. doi: 10.1016/j.earlhumdev.2020.105094. PubMed PMID: 32711341.

6. Fernández de Gamarra-Oca L, Ojeda N, Gómez-Gastiasoro A, Peña J, Ibarretxe-Bilbao N, García-Guerrero MA, et al. Long-Term Neurodevelopmental Outcomes after Moderate and Late Preterm Birth: A Systematic Review. J Pediatr. 2021;237:168-76.e11. Epub 20210624. doi: 10.1016/j.jpeds.2021.06.004. PubMed PMID: 34171360.

7. Machado Júnior LC, Passini Júnior R, Rodrigues Machado Rosa I. Late prematurity: a systematic review. J Pediatr (Rio J). 2014;90(3):221-31. Epub 2014/02/11. doi: 10.1016/j.jped.2013.08.012. PubMed PMID: 24508009.

8. Martínez-Nadal S, Bosch L. Cognitive and Learning Outcomes in Late Preterm Infants at School Age: A Systematic Review. Int J Environ Res Public Health. 2020;18(1). Epub 2020/12/31. doi: 10.3390/ijerph18010074. PubMed PMID: 33374182; PubMed Central PMCID: PMCPMC7795904.

9. McGowan JE, Alderdice FA, Holmes VA, Johnston L. Early Childhood Development of Late-Preterm Infants: A Systematic Review. Pediatrics. 2011;127(6):1111. doi: 10.1542/peds.2010-2257.

10. Srinivasjois R, Silva D, Maina K. Preschool outcomes of late preterm infants-a systematic review. 2014.

11. Teune MJ, Bakhuizen S, Gyamfi Bannerman C, Opmeer BC, van Kaam AH, van Wassenaer AG, et al. A systematic review of severe morbidity in infants born late preterm. Am J Obstet Gynecol. 2011;205(4):374.e1-9. Epub 2011/08/26. doi: 10.1016/j.ajog.2011.07.015. PubMed PMID: 21864824.

12. Tripathi T, Dusing SC. Long-term neurodevelopmental outcomes of infants born late preterm: a systematic review. Research and Reports in Neonatology. 2015;5:91-111.

13. Dong Y, Chen SJ, Yu JL. A systematic review and meta-analysis of long-term development of early term infants. Neonatology. 2012;102(3):212-21. Epub 2012/07/21. doi: 10.1159/000338099. PubMed PMID: 22814228.

14. Nielsen TM, Pedersen MV, Milidou I, Glavind J, Henriksen TB. Long‐term cognition and behavior in children born at early term gestation: A systematic review. Acta Obstetricia et Gynecologica Scandinavica. 2019;98(10):1227-34. doi: 10.1111/aogs.13644.

15. Murray SR, Shenkin SD, McIntosh K, Lim J, Grove B, Pell JP, et al. Long term cognitive outcomes of early term (37-38 weeks) and late preterm (34-36 weeks) births: A systematic review. Wellcome Open Res. 2017;2:101-. doi: 10.12688/wellcomeopenres.12783.1. PubMed PMID: 29387801; PubMed Central PMCID: PMCPMC5721566.

16. Chan E, Leong P, Malouf R, Quigley MA. Long-term cognitive and school outcomes of late-preterm and early-term births: a systematic review. Child Care Health Dev. 2016;42(3):297-312. Epub 2016/02/11. doi: 10.1111/cch.12320. PubMed PMID: 26860873.

---

## [Editor Report · Decision Letter 2]

3 Nov 2023

Early childhood trajectories of domain-specific developmental delay and gestational age at birth: An analysis of the All Our Families cohort

PONE-D-23-18682R2

Dear Dr. Stephenson,

We’re pleased to inform you that your manuscript has been judged scientifically suitable for publication and will be formally accepted for publication once it meets all outstanding technical requirements.

Kind regards,

Shaonong Dang, PhD

Academic Editor

PLOS ONE

Additional Editor Comments (optional):

Authors have addressed the comments fully, and the manuscript has been improved much for publication.
---

## [Editor Report · Acceptance letter]

5 Dec 2023

PONE-D-23-18682R2 

Early childhood trajectories of domain-specific developmental delay and gestational age at birth: An analysis of the All Our Families cohort 

Dear Dr. Stephenson:

I'm pleased to inform you that your manuscript has been deemed suitable for publication in PLOS ONE. Congratulations! Your manuscript is now with our production department. 

Kind regards, 

on behalf of

Dr. Shaonong Dang 

Academic Editor

PLOS ONE